# Using Cone-Beam Computed Tomography to Assess Changes in Alveolar Bone Width around Dental Implants at Native and Reconstructed Bone Sites: A Retrospective Cohort Study

**DOI:** 10.3390/jpm11101011

**Published:** 2021-10-08

**Authors:** Kai-Fang Hu, Szu-Wei Lin, Ying-Chu Lin, Jiiang-Huei Jeng, Yu-Ting Huang, Pei-Feng Liu, Ching-Jiunn Tseng, Yu-Hsiang Chou

**Affiliations:** 1Institute of Clinical Medicine, National Yang Ming Chiao Tung University, Taipei 112, Taiwan; kaifang729@yahoo.com.tw; 2Department of Dentistry, Division of Periodontics, Kaohsiung Medical University Hospital, Kaohsiung 80708, Taiwan; 3School of Dentistry, College of Dental Medicine, Kaohsiung Medical University, Kaohsiung 807378, Taiwan; linszuwei001@gmail.com (S.-W.L.); chulin@kmu.edu.tw (Y.-C.L.); jhjeng@kmu.edu.tw (J.-H.J.); 4Department of Dentistry, National Taiwan University Hospital and School of Dentistry, National Taiwan University Medical College, Taipei 106216, Taiwan; 5Department of Dentistry, Kaohsiung Medical University Hospital, Kaohsiung 80708, Taiwan; 6Department of Medical Research, Division of Medical Statistics and Bioinformatics, Kaohsiung Medical University Hospital, Kaohsiung Medical University, Kaohsiung 80708, Taiwan; stakmuh@gmail.com; 7Department of Biomedical Science and Environmental Biology, Kaohsiung Medical University, Kaohsiung 807378, Taiwan; pfliu908203@gmail.com; 8Department of Medical Research, Kaohsiung Medical University Hospital, Kaohsiung 80708, Taiwan; 9Center for Cancer Research, Kaohsiung Medical University, Kaohsiung 807378, Taiwan; 10Institute of Biomedical Sciences, National Sun Yat-sen University, Kaohsiung 804201, Taiwan; 11Department of Medical education and Research, Kaohsiung Veterans General Hospital, Kaohsiung 813779, Taiwan; 12Department of Medical Research, China Medical University Hospital, China Medical University, Taichung 40402, Taiwan

**Keywords:** retrospective cohort study, implant, bone width, native and reconstructed bone, cone-beam computed tomography

## Abstract

The aim of this study was to use a cone-beam computed tomography (CBCT) to assess changes in alveolar bone width around dental implants at native and reconstructed bone sites before and after implant surgery. A total of 99 implant sites from 54 patients with at least two CBCT scans before and after implant surgery during 2010–2019 were assessed in this study. Demographic data, dental treatments and CBCT scans were collected. Horizontal alveolar bone widths around implants at three levels (subcrestal width 1 mm (CW1), subcrestal width 4 mm (CW4), and subcrestal width 7 mm (CW7)) were measured. A *p*-value of < 0.05 indicated statistically significant differences. The initial bone widths (mean ± standard deviation (SD)) at CW1, CW4, and CW7 were 6.98 ± 2.24, 9.97 ± 2.64, and 11.33 ± 3.00 mm, respectively, and the postsurgery widths were 6.83 ± 2.02, 9.58 ± 2.55, and 11.19 ± 2.90 mm, respectively. The change in bone width was 0.15 ± 1.74 mm at CW1, 0.39 ± 1.12 mm at CW4 (*p* = 0.0008), and 0.14 ± 1.05 mm at CW7. A statistically significant change in bone width was observed at only the CW4 level. Compared with those at the native bone sites, the changes in bone width around implants at reconstructed sites did not differ significantly. A significant alveolar bone width resorption was found only at the middle third on CBCT scans. No significant changes in bone width around implants were detected between native and reconstructed bone sites.

## 1. Introduction

Alveolar bone is subject to continual and rapid remodeling associated with tooth eruption and the subsequent functional demands of mastication [1]. The alveolar process is dependent on the teeth as they develop and remodel with their formation and eruption; therefore, the shape, location, and function of the teeth determine the alveolar morphology [2].

The extraction of a tooth initiates a series of reparative processes involving alveolar bone, periodontal ligaments, and the gingiva [3,4]. After dental extraction, the height of the buccal wall decreases, which results in the disappearance of bundle bone [5]. Resorption of the alveolar ridges most often occurs during the first 6 months after tooth extraction [6]. The rate of resorption varies among different individuals and the same individual at different times. Related factors of the resorption rate are divided into anatomic, metabolic, functional, and prosthetic factors [7].

Dimensional alterations lead to a 50% reduction of ridge width during the first year after extraction in the premolars and molars, where two-thirds of the total changes occur within the first 3 months [4]. Regarding bone width and height resorption, a systematic review revealed a width loss of 2.6–4.5 mm and a height loss of 0.4–3.9 mm for healed sockets [8]. A systematic review by Tan et al. reported that the mean amount of ridge resorption during the first 6 months after tooth extraction was 3.79 ± 0.23 mm in the horizontal dimension and 1.24 ± 0.11 mm in the vertical dimension [9]. Further research has indicated that a thin bone wall phenotype showed more vertical bone resorption than that of a thick bone wall phenotype [10].

In edentulous areas, dental implants have become increasingly prominent among dental replacement options. Their increasing prominence is because they not only provide an ideal chewing function to improve the chewing comfort of patients but also afford esthetic outcomes that enhance dental and facial attractiveness and psychological confidence [11,12,13].

Despite all of the aforementioned benefits, some controversies must still be resolved. Specifically, a consensus has not been reached on whether dental implants can prevent alveolar bone resorption.

Studies have suggested that the force of occlusion and orthodontics did not cause bone resorption or increase the bone density around implants in animal models [14,15,16,17]. By contrast, other studies have argued that dental implants are not effective in maintaining the surrounding hard tissue after the extraction; nevertheless, several studies have concluded that alveolar bone remodeling after extraction would not be affected by implant use in either animal models [18,19,20,21] or humans [22,23].

Alveolar bone responses after implantation are major determinants of the functional and esthetic outcomes of prosthodontic restoration. A study reported that two-dimensional (2D) radiographic images are prone to misjudgment caused by differences in imaging angles or tissue overlap [24]. Cone-beam computed tomography (CBCT) is a suitable tool for measuring the three-dimensional (3D) shape of alveolar bone. It can also achieve high accuracy in the measurement of buccal alveolar bone [25].

However, to date, few studies have discussed changes in the alveolar bone width before and after implantation by comparing CBCT scans. Accordingly, the aim of this study was to measure changes in alveolar bone width before and after dental implant treatment at native and reconstructed (ridge preserved/augmented) bone sites by using CBCT images. Two null hypotheses (*H*_0_) were in this study: (1) no relationship between the bone width and implant placement; (2) no relationship between the change of bone width after implantation and kinds of bone where implants were located.

## 2. Materials and Methods

### 2.1. Study Population

The study protocol (KMUHIRB-E[I]-20200275) was approved by the Institutional Review Board of Kaohsiung Medical University Hospital (KMUH) and conformed to recognized standards of the Declaration of Helsinki.

We enrolled patients who received dental implant treatment at the Division of Periodontics, KMUH, and had at least two CBCT scans of the same tooth sites (i.e., one before the implant surgery and one after it) during 2010–2019. Data on demographic characteristics and dental treatment history, including gender, age at implant surgery, follow-up period, implant position (tooth site and dental arch), implant brand, implant lengths and diameters, and preimplant surgery at implant sites (ridge augmentation or preservation), were collected.

The inclusion criteria were as follows: being aged older than 20 years and having at least one dental implant. The exclusion criteria were as follows: receiving oral and high-dose intravenous administration of bisphosphonates, previous radiographic therapy at the mouth, and having poorly controlled diabetes mellitus (DM). The exclusion criteria for CBCT scans included the following: lacking clarity, lacking anatomic references (e.g., maxillary sinus floor and inferior alveolar nerve canal), or being taken less than 1 month after the implant surgery (Figure 1).

### 2.2. Measurement of the Alveolar Bone Width

To ensure measurement consistency, the change in bone width for each implant between two CBCT scans was measured by a single well-trained examiner (S-W Lin). The CBCT images were captured using Picasso Trio (Vatech, Samsung, Korea) with EzImplant (version 4) viewer software and VGi evo (NewTom, Verona, Italy) with NNT study viewer software. Alveolar bone width measurements were performed using the integral tool of the software. Alveolar bone width measurements were performed using the integral tool of the software. All CBCT images were scanned using Picasso Trio before 2017. Since its launch in early 2017, VGi evo has been used to take most CBCT scans in full mouth reconstruction.

Fixed anatomical landmarks were set in the two CBCT cross-sectional slices of alveolar bone to serve as reference points for presurgery and postsurgery comparisons. Subsequently, the same sites in the two CBCT scans were corrected to ensure that the angles and anatomical features of the two slices matched each other precisely.

The long axes—the vertical distance from the alveolar crest to the floor of the maxillary sinus in the maxilla and that from the alveolar crest to the roof of the inferior alveolar canal in the mandible—were used to measure the three different levels of alveolar bone width before and after surgery at the same height. Because the length distribution of the implants in this study ranged from 8 to 12 mm, measurements were taken at 1, 4, and 7 mm below the bone crest. The horizontal widths of alveolar bone around the implants at the above levels (subcrestal width 1 mm (CW1), subcrestal width 4 mm (CW4), and subcrestal width 7 mm (CW7)) were measured [26,27,28] (Figure 2).

### 2.3. Statistical Analysis

Descriptive data of all patients at baseline were presented as mean ± standard deviation (SD) and frequency distributions to indicate numbers and proportions. The main outcome of the present study was the horizontal widths of alveolar bone around dental implants in two CBCT scans (Figure 3).

A paired *t*-test was applied to detect the difference between the presurgery (baseline) and postsurgery widths at the CW1, CW4, and CW7 levels. A two-sample *t*-test was used to compare changes in bone width around implants between the maxilla and mandible and between native and reconstructed bone. Statistical significance was set at *p* < 0.05. All statistical analyses were conducted using JMP (version 13) software (SAS Institutes Inc., Cary, NC, USA).

## 3. Results

A previous study [29] reported a mean (SD) crestal bone width change of 1.2 (1.5) mm. By considering a 1 mm improvement as clinically important, we determined that 52 participants would be required in each group to yield a power of 0.80 for detecting such a difference using Student’s *t*-test for independent samples in a two-tailed test with α = 0.05. Moreover, we expected that at least 70% of the enrolled patients would complete the study. Accordingly, we calculated that at least a total of 75 patients would be ideal for recruitment in this study.

By using the intraclass correlation coefficient, we conducted a measurement reliability assessment on 34 CBCT images obtained from 24 randomly selected patients. The same examiner repeated the measurements 2 weeks after the initial measurements. The Cronbach’s α value was 0.998, indicating high internal consistency.

A total of 164 implant sites were initially included in this study. Of these, 99 implant sites from 54 patients were considered measurable on the basis of the inclusion and exclusion criteria, and the corresponding patients were included in this study. The demographic data of the recruited patients and data regarding the implants are listed in Table 1.

Fifty-six implants were collected from female patients (56.6%), and the mean age of the patients who received implant surgery was 52.5 ± 11.4 years. The follow-up period after implant surgery was 42.3 ± 32.3 months. Most implants were in the premolars (*n* = 27) and molars (*n* = 66). Half of the implants were in the mandible (*n* = 51). The implants included in this study were of four brands: Straumann^®^ (n = 63), MIS (n = 22), Biomet-3i ^TM^ (n = 13), and Astra Tech (*n* = 1). The length of most implants was 10 mm (*n* = 76; 76.8%) or 11–12 mm (*n* = 20; 20.2%). The implant widths were 3.25 to 3.75 mm (*n* = 20; 20.2%), 4 to 4.2 mm (*n* = 61; 61.6%), and 4.8 to 5 mm (*n* = 18; 18.2%). Forty-eight implants were located at teeth sites treated with alveolar ridge preservation (ARP) or augmentation (48.5%) (Table 1).

Table 2 presents the bone widths around dental implants at three different levels (CW1, CW4, and CW7) in presurgery and postsurgery CBCT scans. In the presurgery CBCT scans, the widths at CW1, CW4, and CW7 were 6.98 ± 2.24, 9.97 ± 2.64, and 11.33 ± 3.00 mm, respectively. In the postsurgery CBCT scans, the widths at CW1, CW4, and CW7 were 6.83 ± 2.02, 9.58 ± 2.55, and 11.19 ± 2.90 mm, respectively. The change in bone width between the presurgery and postsurgery scans was 0.15 ± 1.74 mm at CW1 (*p* = 0.390), 0.39 ± 1.12 mm at CW4 (*p* = 0.0008), and 0.14 ± 1.05 mm at CW7 (*p* = 0.201). Hence, the change in bone width was statistically significant at only the CW4 level (Figure 4).

Table 3 lists the changes in bone width around the implants at the three levels in the maxilla and mandible. In the maxilla, the changes in bone width around the implants at CW1, CW4, and CW7 were 0.10 ± 2.05, 0.53 ± 1.17, and 0.29 ± 1.30 mm, respectively. In the mandible, the changes in bone width around the implants at CW1, CW4, and CW7 were 0.19 ± 1.40, 0.26 ± 1.07, and 0.00 ± 0.74 mm, respectively. No statistically significant difference in the changes in bone width at the three levels was observed between the maxilla and mandible.

Table 4 presents the changes of bone width around the implants at native and reconstructed (ridge preserved/augmented) bone sites. At the reconstructed bone sites, the changes in bone width around the implants at CW1, CW4, and CW7 were −0.04 ± 1.72, 0.21 ± 1.17, and 0.01 ± 1.12 mm, respectively. At the native bone sites, the changes in bone width around the implants at CW1, CW4, and CW7 were 0.34 ± 1.75, 0.56 ± 1.06, and 0.25 ± 0.99 mm, respectively. Compared with the native bone sites, the changes in bone width at the three levels that received reconstructive treatment were not significantly different.

## 4. Discussion

We tried to analyze changes in alveolar width around dental implants at native and reconstructed bones using CBCT. By comparing CBCT images captured before and after implant surgery, we noted a significant bone width change (0.39 ± 1.12 mm) at only the CW4 level. The bone width around the implant was obviously absorbed in the middle third and more stable in the coronal and apical thirds.

For CW1, the nonsignificant difference could be attributed to the measured sites. The irregular connection at the bone crest and the resorption of the bony width at the coronal side often lead to dehiscence defects at the implant platform, thus causing scattering in CBCT images and leading to the misjudgment of the alveolar bone width at the coronal third. Therefore, in this study, cases involving severe bone resorption and difficulties in distinguishing between the implant platform and the bone crest were excluded, and the alveolar bone width at 1 mm below the crest level was measured as the coronal third to reduce measurement errors. However, several studies have reported marginal bone loss around dental implants at the crest level [30,31]. Therefore, although the present study could accurately and clearly determine the alveolar bone width at the crest level, the resorption of coronal bone width around implants might be underestimated.

Regarding the CW7 level, the thicker bone width at the apical third was near the basal bone (i.e., the more stable part of the alveolar bone); hence, the effects engendered by occlusal force and trauma could be excessively small to be observed.

Based on the above finding, the bone width after implantation continuously occurred at three levels. However, the bone width at the CW1 level avoided the crestal marginal bone resorption, and the bone width at the CW7 level near the basal bone was little affected by occlusal loading. Therefore, significant resorption of bone width was only detected at the CW4 level.

When the width and height of the alveolar bone are severely resorbed after tooth extraction, it is usually necessary to strengthen the alveolar ridge to provide the implant with sufficient alveolar bone width. Therefore, many treatment methods and materials for GBR have been proven to significantly increase the width of the alveolar bone [32,33]. Extraction of teeth due to severe periodontitis, the ARP technology can retain more bone and provide sufficient alveolar bone width for the implant [34,35]. The bone width resorption around the implants at the native bone sites was also similar to that around those at reconstructed bone sites. These findings are consistent with those reported by studies on the survival rates and the quantity and quality of bone of implants in native and reconstructed bone [36,37]. Our CBCT findings also demonstrate, for the first time, that the change in bone width around the implants at the native bone sites was similar to that around those at the reconstructed bone sites.

Previous studies have often assessed peri-implantitis or loss of crestal bone level in long-term follow-up periods using periapical or bitewing radiographs. However, periapical and bitewing radiographs are 2D images, which cannot truly show changes in width. CBCT provides a 3D view of an anatomical structure in relation to the teeth and implants and detects width changes on the buccal and lingual sides [27,28]. The alveolar bone surrounding dental implants often maintains a stable height but exhibits dehiscence or fenestration defects on the buccal or lingual side clinically [38]. Therefore, peri-implantitis, which has not been found in the periapical radiographs and bitewing radiographs, could be detected in CBCT images. Apart from clinical probing and dental radiographs, CBCT is an indispensable tool for detecting peri-implantitis [39,40].

Metal objects of titanium implants lead to the formation of artefacts by hardening the X-ray beam [41,42]. These artefacts affect image quality and make it difficult to evaluate osseointegration and inflammation of dental implants [43,44,45,46]. Metallic artefacts limit the visualization quality of the bone around the implants [47]. Despite the limitation of metallic artifacts, the presence of dental implants did not impact the accuracy of measurements of bone thickness in CBCT [48]. When evaluating whether the implant has bony perforation, CBCT can still provide valuable information [47].

The limitations of this study are as follows. First, fenestration and dehiscence often occur at the buccal bone plate of dental implants. Once such defects occur, correctly interpreting bone margin images is difficult, even when CBCT is used [49]. In dehiscence defects, the true bone width can be determined by subtracting the implant width from the original bone width; this can thus yield the width of the bone wall on the intact side. Accordingly, previous studies have only discussed the occurrence of fenestration and dehiscence instead of correctly assessing the remaining bone width.

Compared with implants at the healed bone, immediate implants have a higher incidence of dehiscence [50]. Hence, this study excluded immediate implants and discussed only changes in bone width around implants at the healed bone sites.

Second, when the alveolar bone is severely damaged or absorbed, the bone width or height may not be able to provide sufficient width and height to facilitate dental implant surgery. Once the implant is in a position where the alveolar bone width or height is insufficient, severe peri-implantitis is highly likely to occur and reduce the success and survival rates of the implant. Therefore, this study excluded cases in which no reconstructive surgery was performed before dental implantation at the severely damaged alveolar bone sites. Third, due to a retrospective study, in some recruited sites, the time interval between extraction and implantation cannot be recorded correctly. The longer the extraction time will lead to the more severe alveolar bone resorption. In addition, whether the width of the alveolar bone before tooth extraction will affect the width of the alveolar bone after implantation is unable to explore from the CBCT before tooth extraction. Therefore, in view of the above limitations, well-designed prospective randomized controlled trials will be necessary to further understand other important factors that affect the width of the alveolar bone after implantation.

## 5. Conclusion

Within the limitations of the sample size in this study, we used CBCT scans to observe long-term changes in the alveolar bone width around dental implants at native and reconstructed bone sites. Significant alveolar bone width resorption was observed at only the middle third of all sites. No significant changes in bone width around implants were detected between the maxilla and mandible or between the native and reconstructed bone sites. Bone resorption at the middle third might lead to peri-implantitis, which could be detected early by CBCT scans.

## Figures and Tables

**Figure 1 jpm-11-01011-f001:**
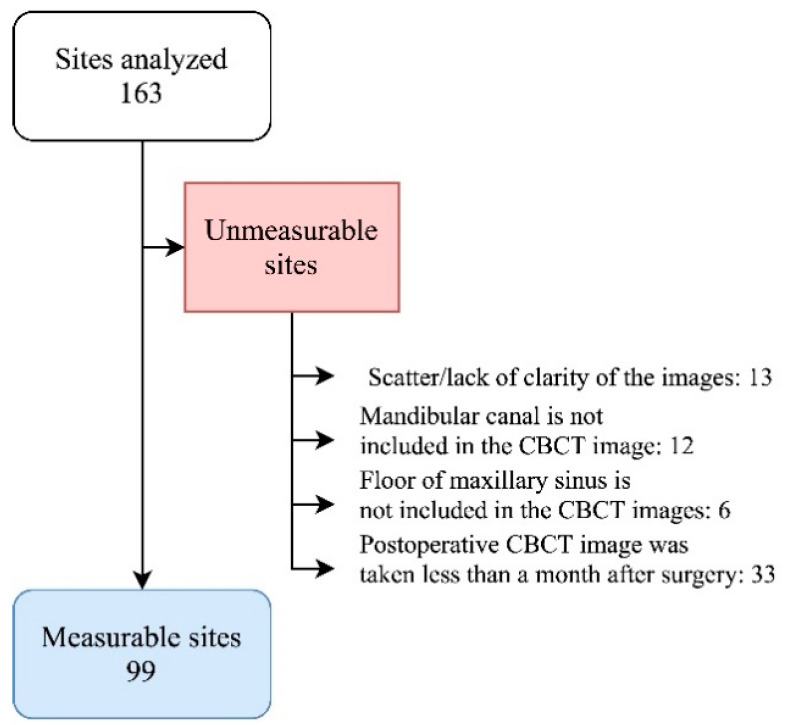
Flow chart of cone-beam computed tomography (CBCT) selection.

**Figure 2 jpm-11-01011-f002:**
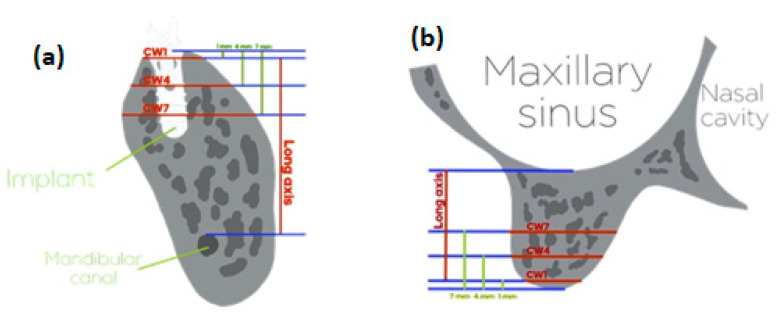
Schematic diagram of the measurement of alveolar bone width at the (**a**) mandible and (**b**) maxilla.

**Figure 3 jpm-11-01011-f003:**
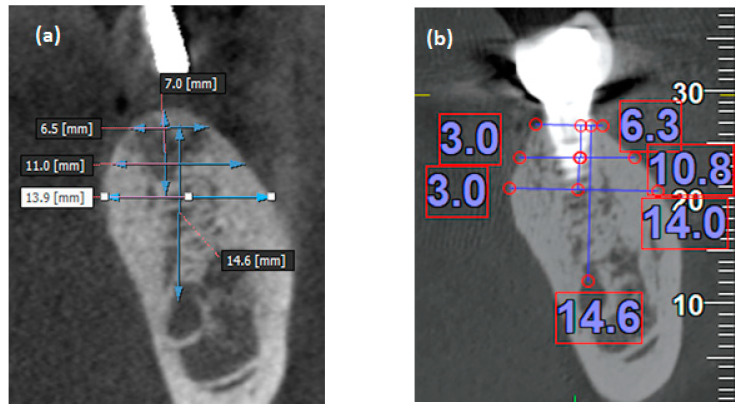
Cross-sectional view of CBCT images demonstrates the way to measure bone width (**a**) before (without implant) and (**b**) after the implant surgery (with implant).

**Figure 4 jpm-11-01011-f004:**
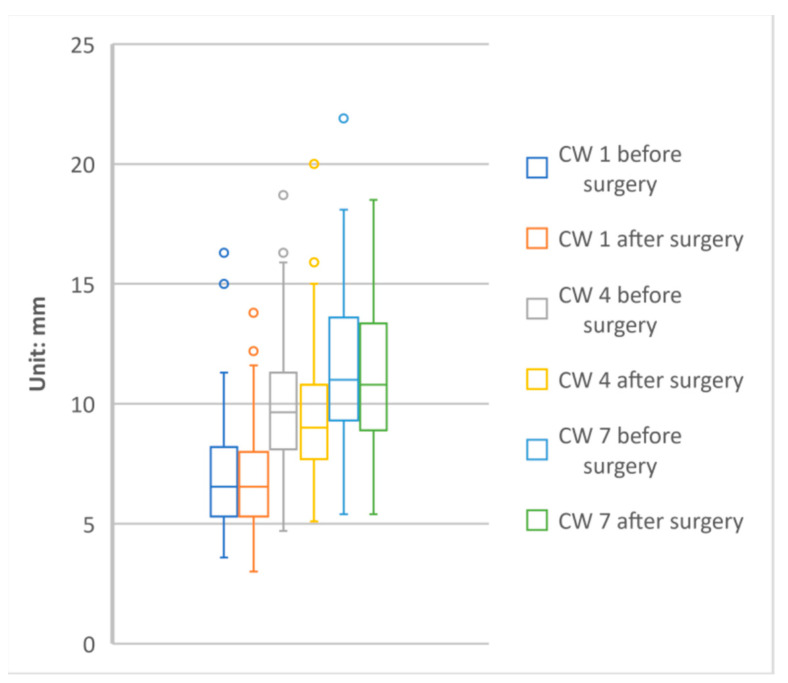
The distributions of alveolar bone width around the implantation site at the three levels before and after surgery; CW: sub-crestal width.

**Table 1 jpm-11-01011-t001:** Demographic data of enrolled patients and dental implants.

Item	*N* (%)
Gender	
Male	43 (43.4)
Female	56 (56.6)
Age of surgery (mean ± SD, years)	52.5 ± 11.4
Follow-up period (mean ± SD, months)	42.4 ± 32.3
Implant sites	
Anterior	6 (6)
Premolar	27 (27.3)
Molar	66 (66.7)
Implant arch	
Maxilla	48 (48.5)
Mandible	51 (51.5)
Brands of implants	
Straumann^®^	63 (63.7)
MIS	22 (22.2)
Biomet 3i^TM^	13 (13.1)
Astra tech	1 (1)
Implant lengths	
8 mm	3 (3)
10 mm	76 (76.8)
11.0–12.0 mm	20 (20.2)
Implant widths	
3.25–3.75 mm	20 (20.2)
4.0–4.2 mm	61 (61.6)
4.8–5.0 mm	18 (18.2)
Previous ridge augmentation or preservation	
Yes	48 (48.5)
No	51 (51.5)

SD: standard deviation.

**Table 2 jpm-11-01011-t002:** The differences in bone widths around implants at three levels were found between before and after the implant surgery.

Sub-Crestal Levels	Bone Widths (mm)			*p*-Value
	Pre-surgery Mean ± SD	Post-surgery Mean ± SD	Change Mean ± SD	
CW 1	6.98 ± 2.24	6.83 ± 2.02	0.15 ± 1.74	0.390
CW 4	9.97 ± 2.64	9.58 ± 2.55	0.39 ± 1.12	0.0008 *
CW 7	11.33 ± 3.00	11.19 ± 2.90	0.14 ± 1.05	0.201

CW: sub-crestal width; SD: standard deviation; *: statistically significant difference as analyzed by paired *t*-test.

**Table 3 jpm-11-01011-t003:** The differences in the change of bone width around implants at three levels were found between maxilla and mandible.

Sub-Crestal Levels	Change of Bone Width (mm)		*p*-Value
	Maxilla Mean ± SD	Mandible Mean ± SD	
CW 1	0.10 ± 2.05	0.19 ± 1.40	0.800
CW 4	0.53 ± 1.17	0.26 ± 1.07	0.226
CW 7	0.29 ± 1.30	0.00 ± 0.74	0.181

CW: sub-crestal width; SD: standard deviation; Statistical analysis by two-sample *t*-test.

**Table 4 jpm-11-01011-t004:** The difference in the change of bone width around implants at three levels were found between implantation at native bones and ridge preservation/augmentation bones.

Sub-Crestal Levels	Change of Bone Width (mm)		*p*-Value
	ARP/GBR Mean ± SD	Native bone Mean ± SD	
CW 1	−0.04 ± 1.72	0.34 ± 1.75	0.272
CW 4	0.21 ± 1.17	0.56 ± 1.06	0.128
CW 7	0.01 ± 1.12	0.25 ± 0.99	0.259

CW: sub-crestal width; SD: standard deviation; ARP: alveolar ridge preservation; GBR: Guided bone regeneration; Statistical analysis by two-sample *t*-test.

## Data Availability

Data available on request due to restrictions eg privacy or ethical. The data presented in this study are available on request from the corresponding author.

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
