# Peer review of "Using Cone-Beam Computed Tomography to Assess Changes in Alveolar Bone Width around Dental Implants at Native and Reconstructed Bone Sites: A Retrospective Cohort Study"

_jpm, 2021, doi:10.3390/jpm11101011_

Round 1

Reviewer 1 Report

I read this manuscript with interest.

The authors emails and reference list must be displayed as per MDPI format.

The title and abstract are clear and provide sufficient information.

The introduction and discussion might be improved. Authors should consider citing the work of Jung, Atwood and others. This will considerably enhance the manuscript and secure more visits. To simplify the authors' search, the reference can be fond below:

  • J Prosthet Dent. 2001 Aug;86(2):119-25. doi: 10.1067/mpr.2001.117609.
  • Evid Based Dent. 2016 Sep;17(3):92-93. doi: 10.1038/sj.ebd.6401193.
  • J N J Dent Assoc. 2015 Spring;86(2):12-3.
  • J Clin Periodontol. 2018 Feb;45(2):265-276. doi: 10.1111/jcpe.12841. Epub 2017 Dec 13.
  • J Clin Periodontol. 2018 Apr;45(4):484-494. doi: 10.1111/jcpe.12860. Epub 2018 Jan 31.
  • Clin Oral Implants Res. 2019 Sep;30(9):872-881. doi: 10.1111/clr.13492. Epub 2019 Jun 24.
  • Clin Oral Implants Res. 2018 May;29(5):522-529. doi: 10.1111/clr.13149. Epub 2018 Apr 1.

Perhaps the authors unintentionally omitted mentioning that this is a retrospective study in the limitations paragraphs.

Author Response

Point 1:

The introduction and discussion might be improved. Authors should consider citing the work of Jung, Atwood and others. This will considerably enhance the manuscript and secure more visits. To simplify the authors' search, the reference can be found below:

J Prosthet Dent. 2001 Aug;86(2):119-25. doi: 10.1067/mpr.2001.117609.

Evid Based Dent. 2016 Sep;17(3):92-93. doi: 10.1038/sj.ebd.6401193.

J N J Dent Assoc. 2015 Spring;86(2):12-3.

J Clin Periodontol. 2018 Feb;45(2):265-276. doi: 10.1111/jcpe.12841. Epub 2017 Dec 13.

J Clin Periodontol. 2018 Apr;45(4):484-494. doi: 10.1111/jcpe.12860. Epub 2018 Jan 31.

Clin Oral Implants Res. 2019 Sep;30(9):872-881. doi: 10.1111/clr.13492. Epub 2019 Jun 24.

Clin Oral Implants Res. 2018 May;29(5):522-529. doi: 10.1111/clr.13149. Epub 2018 Apr 1.

Response 1: Thank you for your suggestion. We had added the statement in line 10-12 in the Introduction section and line 25-30 in the Discussion section.

Point 2: Perhaps the authors unintentionally omitted mentioning that this is a retrospective study in the limitations paragraphs.

Response 2: Thank you for your suggestion. We had added the statement in the last part of limitations paragraphs in the Discussion section.

Reviewer 2 Report

Dear Authors,

The manuscript titled: „Using cone-beam computed tomography to assess changes in alveolar bone width around dental implants at native and reconstructed bone sites: A retrospective cohort study“ has interesting aim,  however I have a few suggestions/questions to clarify parts of the manuscript. Please look at my notes and revise the manuscript accordingly.

The Title of the manuscript well conveys with the major concern of the study.

The Abstract is well written.

The Introduction section well sets up the main topic and introduce the development of the manuscript. However, the null hypothesis is not expressed. The null hypothesis statement must precisely identify the variables assessed through statistical analysis. Please add its statement in the last part of the Introduction section.

In Materials and methods section You mentioned inclusion and exclusion criteria. Why did You choose 20 years as lower age limit if we know that it might be lower or higher than it regarding gender? In exclusion criteria is there other general contraindications except therapy with bisphosphonates or having poorly controlled diabetes? Please clarify this.

Why did you not perform the sample size calculation? Did you base on previously published study?

All measurements were made by one examiner, but please clarify how many measurments were made per one implant?

Could You explain meanings of sentences in line 84-85 and 219-220, because it seems that they are contradictory?

In discussion part You did not analyze main result regarding a significant bone width change at only the CW4 level.

Author Response

Response to Reviewer 2 Comments

Point 1: The Introduction section well sets up the main topic and introduce the development of the manuscript. However, the null hypothesis is not expressed. The null hypothesis statement must precisely identify the variables assessed through statistical analysis. Please add its statement in the last part of the Introduction section.

Response 1: Thank you for your suggestion. We had added the statement in the last part of the Introduction section.

Point 2: In Materials and methods section You mentioned inclusion and exclusion criteria. Why did You choose 20 years as lower age limit if we know that it might be lower or higher than it regarding gender? In exclusion criteria is there other general contraindications except therapy with bisphosphonates or having poorly controlled diabetes? Please clarify this.

Response 2: Thank you for your suggestion. While every teen is different, the growth of the jaw typically continues until 18, 19 or 20 years of age. In some cases, skeletal maturity is finally complete when they are in their mid-20’s. For this reason, we recruited patients receiving implants after 20 years old (not shown in text).

Patients who previously received radiographic therapy at mouth were also excluded in this study. We had added the statement in the Materials and Methods section.

Point 3: Why did you not perform the sample size calculation? Did you base on previously published study?

Response 3: Thank you for your suggestion. We had already mentioned the statement in the first paragraph of Results section.

Point 4: All measurements were made by one examiner, but please clarify how many measurments were made per one implant?

Response 4: After reliability assessment of initial data, every measurement was performed one time per one implant (not shown in text).

Point 5: Could You explain meanings of sentences in line 84-85 and 219-220, because it seems that they are contradictory?

Response 5: Thank you for your suggestion. We had corrected the statement in the first paragraph of Discussion section.

Point 6: In discussion part You did not analyze main result regarding a significant bone width change at only the CW4 level.

Response 6: Thank you for your suggestion. We had added the statement in the line 20 of Discussion section.

Round 2

Reviewer 2 Report

Dear Authors,

After carefully reviewing the revised manuscript titled " Using cone-beam computed tomography to assess changes in alveolar bone width around dental implants at native and reconstructed bone sites: A retrospective cohort study ", all requested/suggested changes were made or authors explained some parts which were not
understandable. The manuscript is well written, the topic is interesting and meets criteria to be part of this journal.